# Reversal of Myofibroblast Apoptosis Resistance and Collagen Deposition by Phaseoloidin-Induced Autophagy Attenuates Pulmonary Fibrosis

**DOI:** 10.3390/biomedicines13112679

**Published:** 2025-10-31

**Authors:** Siyuan Li, Jiazhen Qian, Lang Deng, Wei Liu, Siyuan Tang, Weixi Xie

**Affiliations:** Xiangya Nursing School, Central South University, Changsha 410013, China; m19507490722@163.com (S.L.); qjzh2019@163.com (J.Q.); dengl036@163.com (L.D.); liuw079@csu.edu.cn (W.L.)

**Keywords:** phaseoloidin, AMPK, autophagy, apoptosis, pulmonary fibrosis

## Abstract

**Background and Objectives**: Myofibroblast apoptosis resistance and excessive extracellular matrix (ECM) deposition are central drivers of the irreversibility of pulmonary fibrosis, and both are mechanistically linked to autophagy impairment. *Phaseoloidin* is a bioactive compound derived from *Entada phaseoloides*. This study aimed to investigate the therapeutic potential of *Phaseoloidin* in bleomycin-induced pulmonary fibrosis and its underlying mechanisms. **Methods**:In vivo, the antifibrotic effects of *Phaseoloidin* were evaluated using a bleomycin-induced pulmonary fibrosis mouse model in male C57/BL mice. To further elucidate the mechanisms by which *Phaseoloidin* counteracts fibrosis, in vitro experiments were conducted using primary lung fibroblasts. **Results**: In vitro experiments showed that *Phaseoloidin* could activate the AMPK/mTOR pathway in autophagy-deficient myofibroblasts, effectively reversing autophagic defects and promoting collagen degradation. This autophagy activation selectively degraded PTPN13, a negative regulator of apoptosis, thereby enhancing the sensitivity of myofibroblasts to FasL-induced apoptosis and further facilitating fibrosis resolution. After AMPK gene knockout, the pro-autophagic effect of *Phaseoloidin* completely disappeared, and both collagen clearance and apoptosis recovery were blocked. In vivo experiments confirmed that *Phaseoloidin* exerted antifibrotic effects by activating AMPK-mediated autophagy in myofibroblasts, which significantly ameliorated pulmonary fibrosis. **Conclusions**: *Phaseoloidin* exerts a dual mechanism by activating AMPK-mediated autophagy in myofibroblasts: first, degrading PTPN13 to reverse myofibroblast apoptosis resistance; second, enhancing ECM turnover. These findings indicate that *Phaseoloidin* is a promising novel therapeutic candidate for pulmonary fibrosis.

## 1. Introduction

Idiopathic pulmonary fibrosis (IPF) is a pathological disease characterized by a massive accumulation of myofibroblasts and extensive extracellular matrix (ECM) deposition in lung tissue. Importantly, it is associated with various cellular, biomechanical, and biochemical factors [1]. Due to the insidious nature of pulmonary fibrosis disease, patients have typically already developed severe fibrosis foci by the time of consultation [2]. Previous extensive studies focused on the early prevention of fibrosis in animal models, and most of their mechanisms were found to inhibit the formation of fibrosis foci. There is less research on promoting the regression of established fibrosis foci. Other than lung transplantation, effective treatments for persistent pulmonary fibrosis foci are still lacking, and the corresponding prognosis is poor [3]. Hence, it is critical to explore the pathophysiological mechanisms that promote the regression of pulmonary fibrosis foci via screening and developing new drugs for treating IPF [4].

Studies have shown that the autophagic and apoptotic states of lung fibroblasts are tightly correlated with fibrosis regression [5]. In lung fibrosis, myofibroblasts are characterized by autophagy defects and resistance to apoptosis [6]. Autophagy is a cellular process in which intracellular substances are transported to lysosomes for degradation. It plays a role in the development and aging of organisms by removing excess or damaged organelles, thereby maintaining metabolic and internal environmental homeostasis [7]. Notably, IPF patients exhibit lower-than-normal autophagic activity in lung tissue [8]. In addition, TGF-β treatment decreases the expression of LC3B, an autophagy marker in lung fibroblasts, and reduces autophagic activity [9]. Furthermore, myofibroblasts with autophagy defects cannot properly clear the ECM, a key component of fibrotic foci, resulting in excessive ECM accumulation and accelerated fibrosis.

Apoptosis is an autonomous, genetically controlled, orderly death that maintains homeostasis in vivo [6]. Studies have shown that in IPF patients’ lung tissue, apoptotic stimuli are more frequent and potent, exhibiting, e.g., increased expression of apoptosis-related factors like FasL, but these patients also exhibit decreased apoptosis in myofibroblasts [10]. This phenomenon is mainly associated with myofibroblasts’ reduced sensitivity to apoptosis, focusing on the FasL/Fas signaling pathway [10,11,12]. Myofibroblasts’ apoptosis-resistant state, which corresponds to fibrotic foci, is inextricably linked to refractory fibrosis. Therefore, drugs that target the regulation of autophagy and/or apoptosis may reverse established pulmonary fibrosis.

AMPK (adenosine 5′-monophosphate-activated protein kinase) is a recognized cellular bioenergetic sensor and metabolic regulator consisting of α, β, and γ subunits. It belongs to the serine/threonine (Ser/Thr) kinase family and is widely expressed in various organs [13]. It is increasingly appreciated that AMPK activation in myofibroblasts facilitates reversal of autophagy inhibition and reverses resistance to apoptosis, thereby promoting regression of lung fibrosis [14,15,16].

*Entada phaseoloides* is a traditional Chinese medicinal herb that has been used historically for the treatment of abdominal pain, diabetes, edema, and traumatic arthritis. Phaseoloidin is one of the components of *E. phaseoloides*, exhibiting significant anti-inflammatory and antioxidant activity [17,18,19]. This study reveals that phaseoloidin promotes autophagy in myofibroblasts through activating AMPK, thereby reducing collagen deposition in pulmonary fibrosis, and exerts anti-pulmonary-fibrosis effects by degrading PTPN13 and relieving myofibroblasts of apoptotic resistance to FasL.

## 2. Materials and Methods

### 2.1. Experimental Animals

Regarding the first batch of animals, 75 male C57/BL mice were purchased from the Animal Centre of Central South University, Changsha, China. All the mice acclimated and were free fed over the course of one week. After being anesthetized with sodium pentobarbital, the mice received an intratracheal injection of 50 μL of bleomycin (3 mg/kg) (Nippon Kayaku, Tokyo, Japan) or saline on day 0. Phaseoloidin (Selleck, Ningbo, China) was diluted in a vehicle containing 1% (*v*/*v*) dimethyl sulfoxide (DMSO) in saline; both the vehicle and the phaseoloidin solution were injected intraperitoneally. For the dosing regimen, 75 mice, randomly divided into 5 groups, were treated as follows: (1) control group—saline (*i.t.*) on day 0 + saline containing 1% DMSO (*i.p.*) on days 14–28; (2) BLM group—bleomycin (*i.t.*) on day 0 + saline containing 1% DMSO (*i.p.*) on days 14–28; (3) BLM + 1 mg/kg group—bleomycin (*i.t.*) on day 0 + Phaseoloidin (1 mg/kg, *i.p.*) on days 14–28; (4) BLM + 5 mg/kg group—bleomycin (*i.t.*) on day 0 + Phaseoloidin (5 mg/kg, *i.p.*) on days 14–28; and (5) BLM + 25 mg/kg group—bleomycin (*i.t.*) on day 0 + Phaseoloidin (25 mg/kg, *i.p.*) on days 14–28.

Regarding the second batch of animals, 60 mice were randomly divided into 4 groups and treated as follows: (1) control group—saline (*i.t*.) on day 0 + saline containing 1% DMSO (*i.p*.) on days 14–28; (2) BLM group—bleomycin (*i.t*.) on day 0 + saline containing 1% DMSO (*i.p*.) on days 14–28; (3) BLM + Phaseoloidin group—bleomycin (*i.t*.) on day 0 + Phaseoloidin (25 mg/kg, *i.p*.) on days 14–28; and (4) BLM + Phaseoloidin + CC group—bleomycin (*i.t*.) on day 0 + Phaseoloidin (25 mg/kg, *i.p*.) + Compound C (30 mg/kg, *i.p*.) on days 14–28. The mice were euthanized on day 28 for the collection of lung tissues. The animal experiments complied with the Animal Ethics Committee’s standards and were performed as per the NIH Guide for the Care and Use of Laboratory Animals.

### 2.2. Histological Analysis of Lung Tissues

Lung tissue was fixed with 4% (*w*/*v*) paraformaldehyde, embedded in paraffin, and sectioned. The sections were stained with hematoxylin–eosin and Masson stain.

### 2.3. Immunohistochemistry

Xylene was used to dewax the tissue sections, which were later hydrated in ethanol. The sections were incubated with microwave-extracted antigen, followed by goat serum. Subsequently, they were incubated overnight at 4 °C with collagen I polyclonal antibody (1:200, Millipore, Billerica, MA, USA) and α-SMA monoclonal antibody (1:500, CST, Danvers, MA, USA). After incubation with goat anti-rabbit immunoglobulin G monoclonal antibody (1:5000, SAB, Shanghai, China) for 1 h at room temperature, the sections were counterstained with hematoxylin via incubation with 0.05% diaminobenzidine and rinsed with PBS.

### 2.4. Cell Culture

Lung tissues were shredded and treated with collagenase type I in a 37 °C water bath for 60 min. They were then inoculated into culture flasks for 2–3 generations after being passed through 70 μm and 40 μm filters to obtain purified primary lung fibroblasts and cultured using Dulbecco’s modified Eagle medium (DMEM)-high glucose (Gibco, Grand Island, NY, USA) containing 10% fetal bovine serum (NBS, Sigma, St. Louis, MO, USA), and cells were cultured at 37 °C in a humidified atmosphere containing 5% CO_2_. Myofibroblast induction was conducted by applying TGF-β1 (Peprotech, Cranbury, NJ, USA)-treated lung fibroblasts for 48 h.

### 2.5. Total RNA Preparation and Quantitative Real-Time-Polymerase Chain Reaction (q-PCR)

Trizol (Thermo Fisher Scientific, Waltham, MA, USA) was used to extract RNA from lung tissues and cells. The extracted RNA was reverse-transcribed to cDNA using a reverse transcription kit (Thermo Fisher Scientific, USA) according to the manufacturer’s protocol; SYBR GREEN (Promega, Madison, WI, USA) and a Bio-Rad CFX96 Touch Real-Time PCR Detection System (Bio-Rad, Hercules, CA, USA) were then employed for real-time qPCR. The PCR conditions were as follows: 95 °C for 2 min, with 40 cycles performed; 95 °C for 3 s; and 60 °C for 30 s, followed by a melting curve of 60–95 °C. All primers were commercially synthesized by Sangong Biotechnology Co. (Shanghai, China), and their sequences are provided below:

m-Colla1, F-ATGTTCAGCTTTGTGGACCTC, R-CTGTACGCAGGTGATTGGTG

m-α-SMA, F-TGGCTATTCAGGCTGTGCTGTC, and R-CAATCTCACGCTCGGCAGTAGT.

### 2.6. Western Blotting Analysis

Total lung tissue and frontal cell proteins were extracted using RIPA lysis buffer (Solarbio, Beijing, China), and protein concentrations were detected using bicinchoninic acid (BCA). A total of 10% of the proteins electrophoresed on sodium dodecyl sulfate-polyacrylamide gel electrophoresis gels were transferred to PVDF membranes (Bio-Rad) and 5% (*w*/*v*) skim milk with TBS buffer containing 0.1% (*v*/*v*) Tween 20 (TBST) TBS buffer for blocking. β-ACTIN monoclonal antibody (1:5000, Proteintech, Wuhan, China), α-SMA monoclonal antibody (1:1000, CST, USA), Caspase3 polyclonal antibody (1:1000, Proteintech, China), collagen I polyclonal antibody (1:1000, Millipore, USA), Bcl-2 polyclonal antibody (1:1000, Proteintech, China), BAX polyclonal antibody (1:1000, Proteintech, China), PTPN13 polyclonal antibody (1:1000, RD, Minneapolis, MN, USA), p70S6K polyclonal antibody (1:1000, CST, USA), elf-4B polyclonal antibody (1:500, ABclonal, Wuhan, China), mTOR monoclonal antibody (1:1000, CST, USA), Pho-mTOR monoclonal antibody (1:1000, CST, USA), AMPK monoclonal antibody (1:1000, CST, USA) and Pho-AMPK monoclonal antibody (1:1000, CST, USA) were incubated overnight at 4 °C. The membranes were subsequently washed three times with TBST and incubated with horseradish-peroxidase-conjugated goat anti-rabbit immunoglobulin G monoclonal antibody (1:5000, SAB, China) or mouse anti-goat immunoglobulin G monoclonal antibody (1:5000, SAB, China) or for 2 h at room temperature with Luminata™ Crescendo chemiluminescent horseradish peroxidase substrate (Millipore, USA) to visualize the bands. Furthermore, samples were scanned using a GeneGnome XRQ imager (Syngene, Cambridge, UK).

### 2.7. Hydroxyproline Assay

Hydroxyproline content in lung tissue was measured using a commercial Hydroxyproline Assay Kit (Nanjing Jiancheng Institute of Biotechnology, Nanjing, China). Lungs were excised, rinsed with cold saline, homogenized, adjusted to neutral pH, and diluted to 10 mL. After treatment with activated charcoal and centrifugation, the supernatant was collected; mixed sequentially with reagents R1, R2, and R3; and incubated under the manufacturer’s recommended conditions. Following a final centrifugation at 3500× *g* for 10 min, absorbance was recorded at 550 nm with a spectrophotometer (1 cm optical path). Hydroxyproline concentrations were determined from a standard curve and normalized to wet lung tissue weight.

### 2.8. Immunofluorescence

Expression of Pho-AMPK in lung tissue was detected using immunofluorescence. Sections were permeabilized using 0.5% (*v*/*v*) Triton X-100 (Solarbio). After incubation with goat serum blocking solution (ZSGB Biologicals, Beijing, China), cells were incubated with Pho-AMPK polyclonal antibody (1:100, Proteintech, China) diluted in PBS overnight at 4 °C. After a washing step, the cells were incubated with fluorescent secondary antibody (1:100, Proteintech, China) diluted in PBS containing 0.1% (*v*/*v*) Tween 20 for 1 h at room temperature, stained with DAPI, and observed under a fluorescence microscope.

### 2.9. GFP-LC3B-RFG Plasmids, Vector-PTPN13, and siRNA Transfection

Cells were inoculated in six-well plates under serum-free and antibiotic-free conditions, cultured with antibiotic-free medium for 24 h, and incubated with liposome complexes containing GFP-LC3B-RFP plasmid (HanBio, Shanghai, China), vector-PTPN13 or siRNA (Santa Cruz, Dallas, TX, USA), and Lipofectamine (Invitrogen, Carlsbad, CA, USA) for 24 h for subsequent experimental treatments.

### 2.10. TUNEL Assay

The TUNEL assay was used to detect the number of apoptotic cells using the TUNEL kit (Elabscience, Wuhan, China). After collection and fixed permeabilization, 100 μL of TdT Equilibration Buffer was added dropwise to each sample, and the reaction was carried out at 37 °C for 10–30 min. After aspiration of TdT Equilibration Buffer, 50 μL of labeling solution was added dropwise to each sample, and the reaction was carried out in a wet box at 37 °C for 60 min under low light. The nuclei were stained with DAPI, sealed, and observed under a fluorescence microscope.

### 2.11. Annexin V-FITC-PI Double-Staining Assay (Cell Apoptosis Assay)

The Annexin V-FITC-PI double-staining assay was employed alongside flow cytometry to detect the level of apoptosis. Cells were treated and spread out in 400 μL binding buffer. Then, myofibroblasts were stained with 2 μL Annexin V and 1 μL propidium iodide (PI) (BD Biosciences, Franklin Lakes, NJ, USA) and incubated for 20 min.

### 2.12. Co-Immunoprecipitation Assay

A co-immunoprecipitation assay (Beyotime, Shanghai, China) was used to detect the binding of p62 and PTPN13. Magnetic beads were added to the lysed cells and rotated at 4 °C for 2 h, and the supernatant was collected after the magnetic rack separated the beads. Then, the pretreated bead–antibody complex was added, incubated at 4 °C for 2 h, and washed three times with lysis buffer. Subsequently, the supernatant was discarded, and 20 μL of 2 × SDS loading buffer was added to the magnetic beads, which were boiled at 95 °C for 5 min. Finally, the supernatant was taken as a sample for detection via SDS-PAGE electrophoresis.

### 2.13. Statistical Analysis

All data are expressed as the mean ($\bar{x}$) ± standard deviation and underwent statistical analysis using GraphPad Prism v9.0 software. For data conforming to a normal distribution, one-way analysis of variance was employed to assess group differences. A *p*-value less than 0.05 was considered statistically significant.

## 3. Results

### 3.1. Phaseoloidin Facilitates the Regression of Bleomycin-Induced Lung Fibrosis in Mice

In one study, the synthesis of extracellular matrix proteins, including collagen, and the number of myofibroblasts were reported to increase significantly from day 14 after bleomycin administration, indicating that significant pulmonary fibrosis was already established by day 14 in mice [20]. To investigate the effect of phaseoloidin on induced lung fibrosis, we administered 1 mg/kg, 5 mg/kg, and 25 mg/kg of phaseoloidin to mice for 14 consecutive days intraperitoneally to observe pulmonary fibrosis. Hematoxylin–eosin (HE) and Masson staining showed that phaseoloidin concentrations reduced alveolar wall thickening and extracellular matrix deposition, with an optimal effect at 25 mg/kg (Figure 1A). Immunohistochemical staining revealed similar results for Collagen I and α-SMA, markers of pulmonary fibrosis (Figure 1A). Moreover, H&E and Masson’s staining, along with hydroxyproline quantification, demonstrated that phaseoloidin alone did not induce pulmonary fibrosis or increase collagen deposition relative to the controls (Appendix A). Survival-curve analysis demonstrated that dose-dependent phaseoloidin treatment reduced mortality following bleomycin exposure and partially reversed bleomycin-induced weight loss (Figure 1B,C). The reduction in hydroxyproline content further confirmed the anti-fibrotic effect of phaseoloidin on bleomycin-treated mice (Figure 1D). Furthermore, phaseoloidin reduced bleomycin-induced upregulation of Collagen I and α-SMA mRNA and protein levels (Figure 1E–G). Therefore, phaseoloidin exerts anti-fibrotic effects in bleomycin-treated mice.

### 3.2. AMPK/mTOR Pathway Mediates the Autophagy-Enhancing Effect of Phaseoloidin

Autophagy is a biological process that is essential for the metabolic needs and renewal of organelles [21]. In vivo, phaseoloidin was found to activate AMPK as well as promote autophagy in lung tissue (Figure 2A). To determine whether phaseoloidin treatment could increase autophagy levels in myofibroblasts, myofibroblasts were generated via 48 h of TGF-β induction in primary mouse lung fibroblasts. Following the initial 48 h, another 48 h intervention with 10 μM, 30 μM, and 100 μM phaseoloidin (Figure 2B) was conducted. GFP-LC3B-RFP transfection and WB showed that the phaseoloidin treatment enhanced myofibroblasts’ autophagy levels (Figure 2C,D). Additionally, numerous studies have shown that mTORC1 is a crucial regulator of autophagy [22,23,24]. Furthermore, we examined the protein expression of p70S6K and elf-4B, the latter of which is the main effective downstream protein of mTORC1, representing its activity and regulating the autophagy statuses of cells (Figure 2D). Phaseoloidin levels were found to decrease phosphorylation of mTOR in a concentration gradient. Decreases in p70S6K and elf-4B levels followed, thereby promoting autophagy. Furthermore, phaseoloidin was found to reduce the expression of p62 and enhance the conversion of LC3B I to LC3B II, further supporting its role in facilitating autophagy (Figure 2D). AMPK, a heterotrimeric structure consisting of α, β, and γ subunits, is a pivotal molecule in preventing or delaying the formation of fibrosis and acts as the upstream signaler for the phosphorylation of mTOR(Ser2448) at the Thr172 site [13]. Therefore, we speculate that phaseoloidin may exert autophagy-promoting effects via AMPK. WB showed that phaseoloidin could active AMPK in a concentration gradient (Figure 2E). Moreover, AMPK siRNA reversed the decrease in p70S6K and elf-4B levels following phaseoloidin treatment. Moreover, the results regarding p62 and LC3B were also reversed (Figure 2F). Collectively, phaseoloidin can promote autophagy via the AMPK/mTOR pathway.

### 3.3. Phaseoloidin Promotes Collagen Degradation via Autophagy

Collagen I, a marker of the formation of fibrous foci, is a significant component of the ECM. When a lung injury occurs, collagen can repair the interstitial lung tissue, yet it produces large deposits when cleared abnormally, leading to fibrosis of the lung tissue [25]. A CCK8 assay was used to determine the effect of phaseoloidin on myofibroblast cell viability across five concentration gradients (3, 10, 30, 100, and 300 μM). The results demonstrated that phaseoloidin did not have any cytotoxic effects on myofibroblasts at any of the tested concentrations (Figure 3A). Next, we determined whether phaseoloidin treatment could promote collagen degradation in myofibroblasts. WB indicated that the concentration gradient of phaseoloidin reduced the abundance of Collagen I and was optimal at a concentration of 100 μM (Figure 3B). Yet, in the presence of CQ and 3-MA—two autophagy inhibitors—phaseoloidin’s effect on Collagen I clearance and LC3B was reversed (Figure 3C), indicating that the promotion of collagen degradation via phaseoloidin occurs through autophagy. After our investigation of phaseoloidin with autophagy inhibitors, AMPK siRNA was used to determine whether phaseoloidin enhanced autophagic collagen clearance via the AMPK/mTOR pathway. Collagen clearance by phaseoloidin was blocked by siRNA (Figure 3D). Therefore, our findings suggest that the autophagic clearance of collagen promoted by phaseoloidin is mediated by the AMPK/mTOR pathway.

### 3.4. Phaseoloidin Resolves Myofibroblast Apoptosis Resistance Induced by Insensitivity to FasL

Myofibroblasts’ apoptosis-resistant properties are the main reason behind the difficulty of fibrosis regression, of which insensitivity to FasL is the key reason [10]. To explore whether this apoptosis resistance induced by insensitivity to FasL could be reversed by phaseoloidin, primary myofibroblasts were treated with phaseoloidin and FasL. TUNEL staining and flow cytometry demonstrated that phaseoloidin reversed myofibroblasts’ apoptosis resistance (Figure 4A,B). Moreover, WB indicated that the abundance of the anti-apoptotic protein Bcl-2 was downregulated following phaseoloidin treatment, while the abundance of pro-apoptotic protein BAX was upregulated, and the level of Cleaved Caspase-3 increased (Figure 4C). Flow cytometry showed that AMPK siRNA blocked the restoration of apoptosis resistance caused by insensitivity to FasL via phaseoloidin treatment (Figure 4D). Furthermore, AMPK siRNA reversed the downregulation of Bcl-2 protein, the upregulation of BAX protein, and the upregulation of Cleaved Caspase-3 protein induced by phaseoloidin (Figure 4E). Thus, phaseoloidin reversed myofibroblasts’ apoptosis resistance induced by insensitivity to FasL.


Figure 3Phaseoloidin facilitates collagen autophagic degradation. (**A**) CCK8 assay reflects the toxicity of phaseoloidin glucoside on myofibroblasts. (**B**) Primary mouse fibroblasts became myofibroblasts after induction by TGF-β1 for 48 h. They were treated with 10 μM, 30 μM, and 100 μM of phaseoloidin solution for 48 h. Western blot analysis of Collagen I (**left**) and quantification of the band intensity (**right**). (**C**) Western blot analysis of Collagen I and LC3B in CQ and 3-MA pretreated primary mouse myofibroblasts with 100 μM phaseoloidin (**left**) and quantification of the band intensity (**right**). (**D**) Western blot analysis of Collagen I under Ctl siRNA or AMPK siRNA treated (**left**) and quantification of the band intensity (**right**). Data in this figure represent the means ± S.D. (*ns*, not Significance, * *p* < 0.05, ** *p* < 0.01, *** *p* < 0.001, **** *p* < 0.0001).
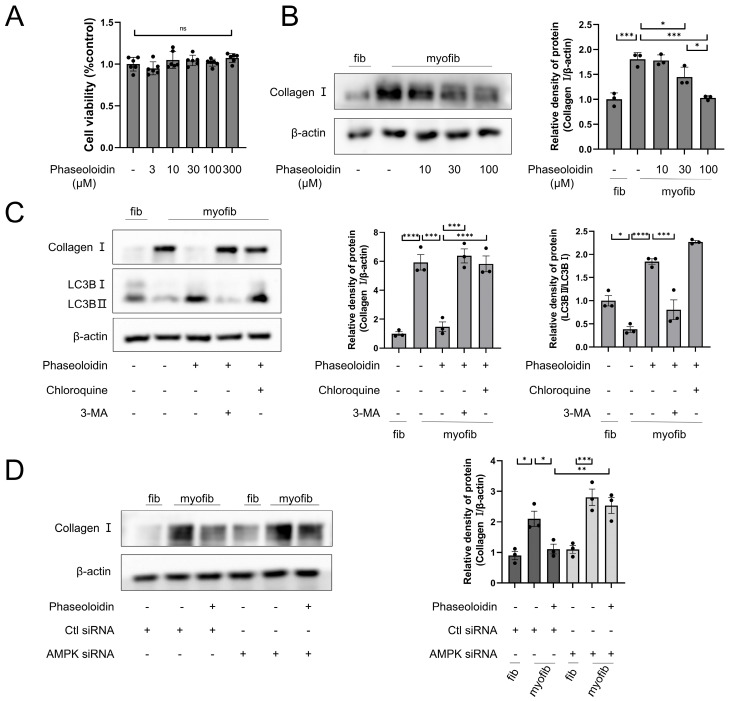

Figure 4Apoptosis resistance of myofibroblasts to FasL is reserved by phaseoloidin. (**A**) TUNEL assay was used to detect apoptosis in primary mouse myofibroblasts after co-treatment with FasL and 10 μM, 30 μM, and 100 μM phaseoloidin for 48 h. Green represents TUNEL, and blue represents nuclei. (**B**) Flow cytometry was used to detect the percentage of apoptosis in the above experimental groups. (**C**) Western blot analysis of Bcl-2, BAX, full-length Caspase3, and Cleaved Caspase3 (**left**) and quantification of the band intensity (**right**). (**D**) Flow cytometry was used to detect the percentage of apoptosis under Ctl siRNA or AMPK siRNA treated. (**E**) Western blot analysis of Bcl-2, BAX, full-length Caspase3, and Cleaved Caspase3. Bars represent 20 μm (**A**). Data represent the means ± S.D. (* *p* < 0.05, ** *p* < 0.01, *** *p* < 0.001, **** *p* < 0.0001).
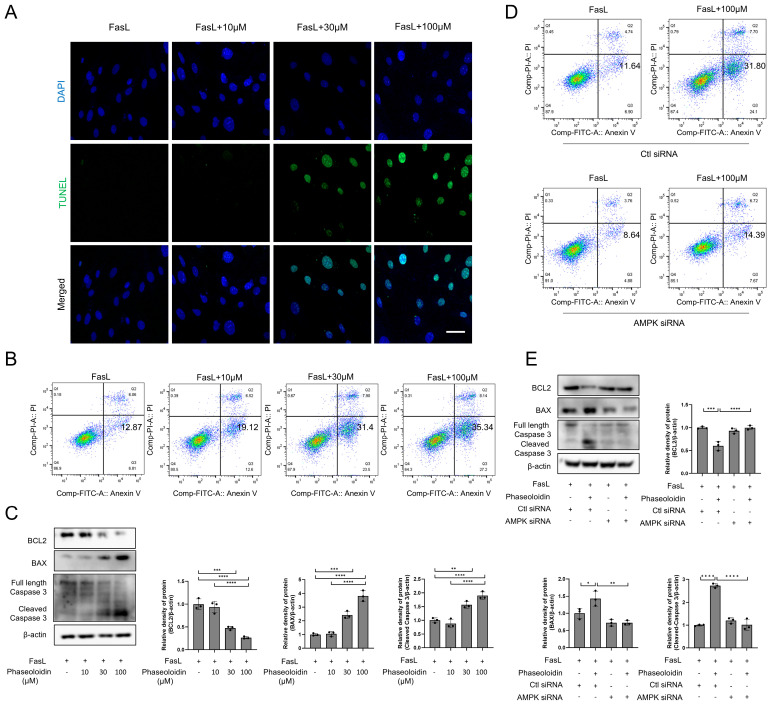



### 3.5. PTPN13 Is Linked to the Effects of Phaseoloidin on Autophagy and Apoptosis

Numerous studies have shown the complex interactions that occur between autophagy and apoptosis, with classical apoptotic factors regulating autophagy and mediating apoptosis [26,27,28]. PTPN13 is a cytoplasmic tyrosine phosphatase that is associated with the Fas receptor and inhibits the assembly of the death-inducing signaling complex (DISC), thereby suppressing FasL-induced extrinsic apoptosis [29,30]. Ubiquitinated PTPN13 is selectively recognized by the autophagy receptor p62 and targeted by autophagosomes for lysosomal degradation, which increases Fas phosphorylation and sensitizes cells to FasL-induced apoptosis [30]. First, we examined the changes in PTPN13 protein levels that occurred during co-treatment of phaseoloidin and FasL. We found that treatment with a gradient of phaseoloidin concentrations reduced the protein abundance of PTPN13 (Figure 5A). p62, as a junction protein of PTPN13, is a vital regulator of PTPN13 degradation. CO-IP results showed that treatment with phaseoloidin promoted the binding of PTPN13 and p62 (Figure 5B). It has been hypothesized that phaseoloidin enhances autophagy by promoting the degradation of PTPN13 and p62 conjugates after recognition by autophagosomes. To further demonstrate that the restoration of apoptosis resistance (caused by insensitivity to FasL) triggered by phaseoloidin was mediated via autophagy, CQ was treated 2 h earlier than phaseoloidin and FasL. Notably, the results showed that CQ-mediated autophagy inhibition reversed the effect of phaseoloidin on apoptosis (Figure 5B–D). Furthermore, we noted a blockade of apoptosis by phaseoloidin after overexpressing PTPN13 in myofibroblasts (Figure 5E). Taken together, the results show that phaseoloidin restored apoptosis resistance induced by insensitivity to FasL by increasing autophagy to promote the degradation of PTPN13.

### 3.6. AMPK Mediates the Anti-Fibrotic Effect of Phaseoloidin on Bleomycin-Induced Pulmonary Fibrosis in Mice

After the anti-fibrotic effect of phaseoloidin via AMPK in vitro had been confirmed, we validated the AMPK-mediated action of phaseoloidin in vivo. We injected mice with bleomycin-induced pulmonary fibrosis with 30 mg/kg of Compound C, a specific inhibitor of AMPK, 2 h before injecting them with phaseoloidin. HE and Masson staining were employed. The immunohistochemistry pertaining to Collage I and α-SMA showed that Compound C reversed phaseoloidin’s anti-fibrotic effect (Figure 6A). We also examined the immunofluorescence staining of Pho-AMPK at the Thr172 site and found that the phaseoloidin treatment enhanced the phosphorylation level of Thr172, for which these effects were blocked by compound C (Figure 6A). Compound C also reversed the increased survival and weight gain in the phaseoloidin-treated mice (Figure 6B,C). Furthermore, qPCR and WB results similarly demonstrated that compound C blocked the activation of AMPK and the anti-fibrotic effect of phaseoloidin (Figure 6D–F). Therefore, AMPK mediates the antifibrotic effect of phaseoloidin in vivo.

**Figure 5 biomedicines-13-02679-f005:**
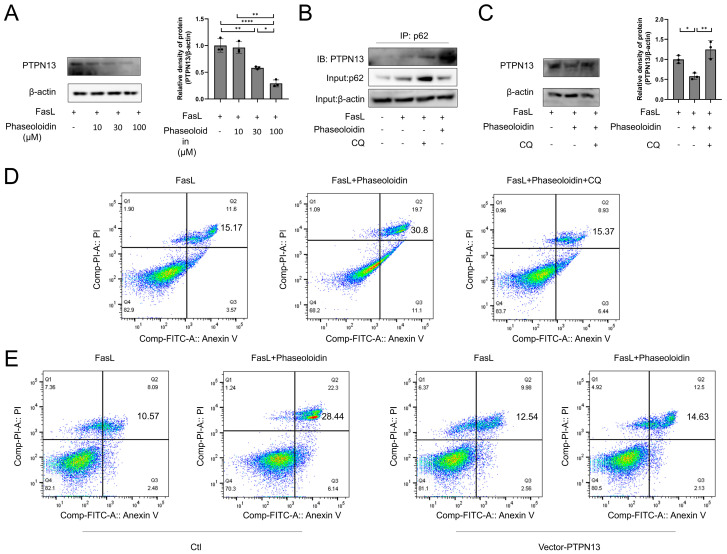
PTPN13 mediated by phaseoloidin links autophagy and apoptosis. (**A**) Western blot analysis of PTPN13 levels in FasL and different dose phaseoloidin co-treated myofibroblasts (**left**) and quantification of the PTPN13 band intensity (**right**). (**B**) Western blotting reflects the CO-IP assay of PTPN13 binding to p62 under co-treatment with FasL and phaseoloidin or CQ. (**C**) Western blot analysis of PTPN13 levels in FasL and 100 μM phaseoloidin co-treated primary mouse myofibroblasts in the presence or absence of CQ (**top**) and quantification of the PTPN13 band intensity (**bottom**). (**D**) Flow cytometry was used to detect the percentage of apoptosis in the above experimental groups. (**E**) Flow cytometry was used to detect the percentage of apoptosis in groups with or without vector-PTPN13. Data in this figure represent the means ± S.D. (* *p* < 0.05, ** *p* < 0.01, **** *p* < 0.0001).

**Figure 6 biomedicines-13-02679-f006:**
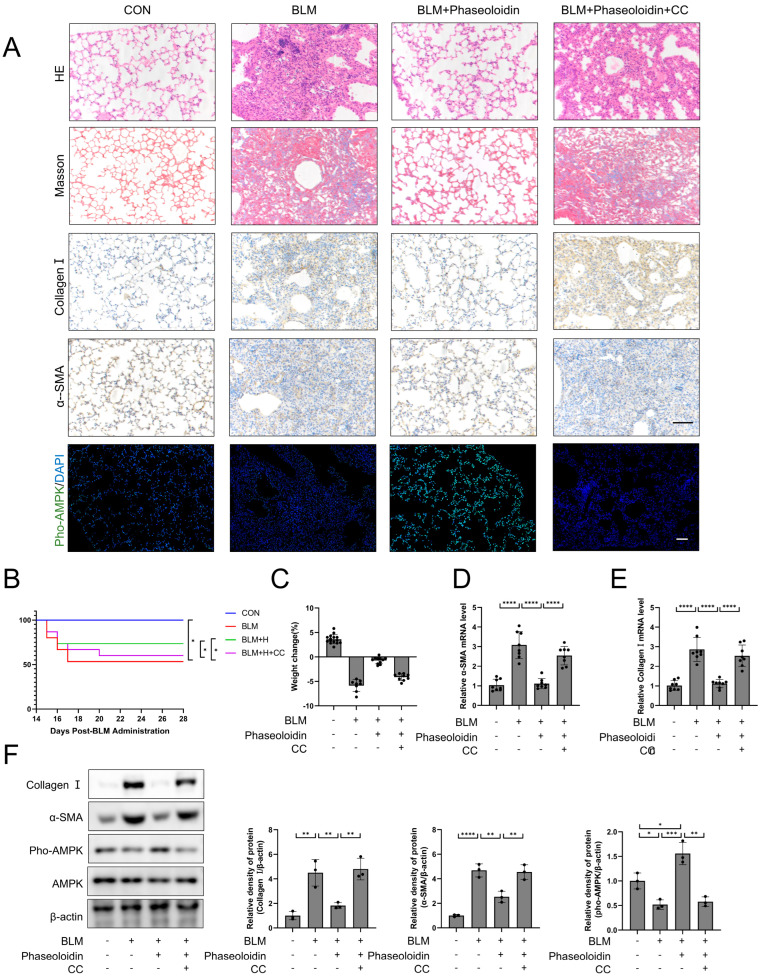
The antifibrotic effect of phaseoloidin in bleomycin-induced pulmonary fibrosis in mice is regulated by AMPK. (**A**) Bleomycin (5 mg/kg) was given by tracheal injection on day 0, and phaseoloidin (30 mg/kg) was given before phaseoloidin (25 mg/kg) by intraperitoneal injection from day 14 to day 28. Histology, Masson staining, immunohistochemistry of Collagen I and α-SMA, and immunofluorescence of Pho-AMPK were analyzed for control, BLM, BLM + phaseoloidin, BLM + phaseoloidin + CC groups. (**B**) Survival curves of the above experimental animal groups. (**C**) Weight changes in the above experimental animal groups. (**D**,**E**) Quantification of mRNA expression of α-SMA and Collagen I in the above-mentioned animal experimental groups. (**F**) Western blot analysis of Pho-AMPK, Collagen I, and α-SMA levels in experimental animal groups (**left**) and quantification of the band intensity (**right**). Bars represent 100 μm (**A**). Data represent the means ± S.D. (* *p* < 0.05, ** *p* < 0.01, *** *p* < 0.001, **** *p* < 0.0001).

## 4. Discussion

In a previous study, it was confirmed that phaseoloidin exerts its anti-fibrotic effects by activating AMPK [31]. Additionally, it was demonstrated for the first time, both in vitro and in vivo, that phaseoloidin activates AMPK and enhances autophagy in myofibroblasts with autophagy defects. Notably, this process occurred through the mTOR signaling pathway promoting the clearance of extracellular matrix. Furthermore, the enhanced autophagy relieved myofibroblasts’ PTPN13-mediated resistance to FasL-induced apoptosis, allowing lung fibrosis to subside. These novel results suggest that phaseoloidin could be considered for use as a drug for the treatment of pulmonary fibrosis (Figure 7).

Studies have shown that in a bleomycin-induced pulmonary fibrosis model, the degree of pulmonary fibrosis gradually increases with the suppression of autophagic activity [21]. Autophagy-defective mice have also been shown to develop more severe pulmonary fibrosis after bleomycin treatment than wild-type mice [32,33]. Moreover, autophagy activation is vital for the regression of inflammation and fibrosis after an acute tissue injury [7,34]. Collagen, a marker of the formation of fibrous foci, is a significant component of the ECM. Under compensable conditions, procollagen chains form stable structures that the proteasome degrades via the ERAD pathway [25]. With decompensation, autophagy can be involved in collagen degradation in the ECM via multiple pathways, and autophagic vesicles that can wrap and sequester cytoplasmic components can be formed to enhance the elimination of collagen through the autophagic lysosomal pathway [35,36]. In pulmonary fibrosis, defective autophagy is an important feature of myofibroblasts. In this study, we novelly demonstrated that phaseoloidin significantly attenuates bleomycin-induced pulmonary fibrosis (as shown in Figure 1). In the bleomycin-induced pulmonary fibrosis model, the period from day 1 to day 14 after bleomycin injection is primarily characterized by inflammatory damage, while the period from day 15 to day 28 is marked by fibrosis, featuring extensive extracellular matrix (ECM) deposition [37]. However, in most studies, drug interventions begin on day 1, exerting antifibrotic effects possibly due to their anti-inflammatory properties rather than direct antifibrotic effects [20]. In our study, phaseoloidin was administered from days 15 to 28, and the results in Figure 1 demonstrate that it may have directly alleviated pulmonary fibrosis. Additionally, defective autophagy is an important feature of myofibroblasts in fibrosis [35,36]. Previous studies primarily demonstrated that autophagy enhancement can attenuate pulmonary fibrosis by suppressing the epithelial–mesenchymal transition (EMT) [38]. For instance, FGF21 alleviates fibrosis through mTOR-dephosphorylation-mediated autophagy induction, with its core mechanism centered on EMT inhibition [39]. In contrast, our research focuses on mitigating pulmonary fibrosis by promoting myofibroblast autophagy and collagen degradation. However, the regulatory roles of phaseoloidin in alveolar epithelial cells and macrophages regarding autophagy modulation remain unexplored. This aspect will be systematically investigated in our future research. In conclusion, we suggest that PTPN13 mediates the phaseoloidin-induced restoration of apoptosis resistance in myofibroblasts. Although our study is limited to experimental models and did not include clinical trials, the findings provide important preclinical evidence supporting the therapeutic potential of Phaseoloidin. Future work will focus on validating these effects in human lung tissue or clinical samples, which may offer valuable insights for translating this compound into potential therapeutic strategies.

Apoptosis is an autonomous and orderly death controlled by genes used to sustain homeostasis [40]. During the repair of a lung injury, activated fibroblasts show changes corresponding to apoptosis and gradually disappear [41]. Myofibroblasts exhibit apoptosis resistance at the time of fibrosis, resulting in the continued accumulation of myofibroblasts in pulmonary fibrotic tissue. Research suggests that IPF lung fibroblasts receive a higher frequency and level of apoptotic stimuli than age-matched controls [42]. However, apoptosis is reduced, making cells’ reduced sensitivity to apoptotic signals a compelling explanation [42]. Apoptosis induced by the death receptor Fas, a key player in the control of lung fibroblasts, was shown to be influenced by FasL once a predetermined threshold was exceeded [43,44,45,46]. Conversely, IPF patients with lung fibroblasts showed resistance to FasL-induced apoptosis [11,45], which was highly correlated with the downregulation of Fas expression and increased expression of multiple apoptosis-regulating genes, including Bcl-2, XIAP, cFLIP_L_, and PTPN13 [12,47,48,49]. These findings suggest that lung fibroblasts’ acquired resistance to FasL-induced apoptosis promotes the accumulation of fibroblasts with persistent IPF. Existing studies have demonstrated that reversing apoptosis resistance in myofibroblasts can effectively alleviate pulmonary fibrosis [48]. For instance, Metformin reverses established pulmonary fibrosis by promoting the apoptosis of myofibroblasts, and BTSA1 induces apoptosis in senescent myofibroblasts via targeted activation of BAX [15,50]. Consistent with this picture, as shown in Figure 4, phaseoloidin serves as a trigger that causes FasL-stimulated programmed death of myofibroblasts. This finding not only lays the foundation for our study of phaseoloidin and FasL-mediated apoptosis but also reveals a limitation: we only assessed autophagic flux at the 48 h timepoint of phaseoloidin treatment, without examining its long-term effects on autophagosome–lysosome fusion.

Numerous studies have shown that complex interactions take place between autophagy and apoptosis [26,27,28,51]. Among them, classical apoptosis regulators can regulate autophagy, and autophagy can likewise promote apoptosis [52,53,54]. Furuya et al. found simultaneous upregulation of apoptosis and autophagy in ceramide-treated breast and colon cancer [55]. Fitzwalter et al. explored differences in the responses of cells with different levels of autophagic flux to cell-death stimuli; strikingly, they found that BJAB leukemia cells with high autophagic flux were more sensitive to FasL-induced apoptosis than lower-autophagic-flux cells [56]. PTPN13 plays a critical regulatory role as a link between autophagy and apoptosis. Recent studies have shown that genetic defects in PTP-BL, the mouse homolog of concurrent human PTPN13, reduce bleomycin-induced pulmonary fibrosis in mice [48]. In this study, we found that phaseoloidin could decrease the expression of PTPN13. We demonstrated that phaseoloidin decreases PTPN13 expression through co-degradation of PTPN13 and p62, suggesting that PTPN13 may serve as a critical therapeutic target of phaseoloidin. While we identified PTPN13 as a critical mediator in the phaseoloidin-induced reversal of apoptosis resistance, its precise molecular mechanisms remain incompletely elucidated. Therefore, our data indicate that the phaseoloidin-induced reversal of apoptosis resistance in myofibroblasts may be mediated by PTPN13.

AMPK is a pivotal molecule in preventing or delaying fibrosis [57]. A recent study indicated the role of defective AMPK signaling in persistently activated bone marrow fibroblasts within fibrotic foci [58]. IPF patients have significantly reduced AMPK activity in areas of active fibrosis in the lungs [15]. In preclinical studies, AMPK activators protected against lung injury, including via airway remodeling in asthma, and attenuated the subsequent development of fibrosis [59]. Furthermore, in addition to preventing fibrosis progression, the AMPK activator metformin promoted the regression of established fibrosis in a bleomycin model [15]. AMPK and mTOR form a critical energy-sensing axis regulating autophagy [60]. “Activated AMPK phosphorylates TSC2, which inhibits Rheb GTPase and subsequently suppresses mTORC1”. This mTORC1 inhibition enhances autophagic flux, promoting the clearance of damaged organelles and misfolded proteins, a process essential for cellular homeostasis [60]. In fibrosis, disrupted AMPK/mTOR signaling impairs autophagy, contributing to persistent myofibroblast activation and ECM accumulation [60,61]. Research has demonstrated that mTOR inhibition effectively attenuates pulmonary fibrosis [62]. Moreover, a study demonstrated that tanshinone IIA attenuates pulmonary fibrosis by simultaneously activating AMPK and inhibiting mTOR signaling. Consistent with these findings, our study reveals that phaseoloidin (PHA) promotes AMPK phosphorylation at Thr172 while suppressing mTOR phosphorylation at Ser2448 and the expression of its downstream effector. As shown in Figure 6, the results demonstrated complete blockade of phaseoloidin’s anti-fibrotic activity following AMPK inhibition. In a previous study, it was confirmed that phaseoloidin exerts its anti-fibrotic effects by activating AMPK [31]. Our study is the first to demonstrate that phaseoloidin attenuates pulmonary fibrosis in vivo through AMPK activation. However, the compound’s therapeutic efficacy remains unexplored in other pulmonary injury models beyond the fibrotic context. In conclusion, it is hypothesized that phaseoloidin’s effect in promoting myofibroblasts’ autophagy and restoring their apoptosis resistance state is mediated by the activation of AMPK.

Phaseoloidin, one of the constituents of the traditional medicinal plant *Entada phaseoloides*, has been demonstrated to possess anti-inflammatory and antioxidant properties [17,18,19]. Therefore, we investigated whether phaseoloidin could accelerate fibrosis regression in a bleomycin-induced pulmonary fibrosis model. The in vivo results indicated that phaseoloidin could promote collagen degradation and the elimination of fibrotic foci and attenuate pulmonary fibrosis in mice. In vitro experiments further revealed that phaseoloidin promoted myofibroblast autophagy by activating AMPK while reducing the level of PTPN13 to reverse apoptosis resistance. In conclusion, we have demonstrated that phaseoloidin can facilitate the regression of established fibrosis and may be a valuable drug for treating progressive fibrotic diseases.

## Figures and Tables

**Figure 1 biomedicines-13-02679-f001:**
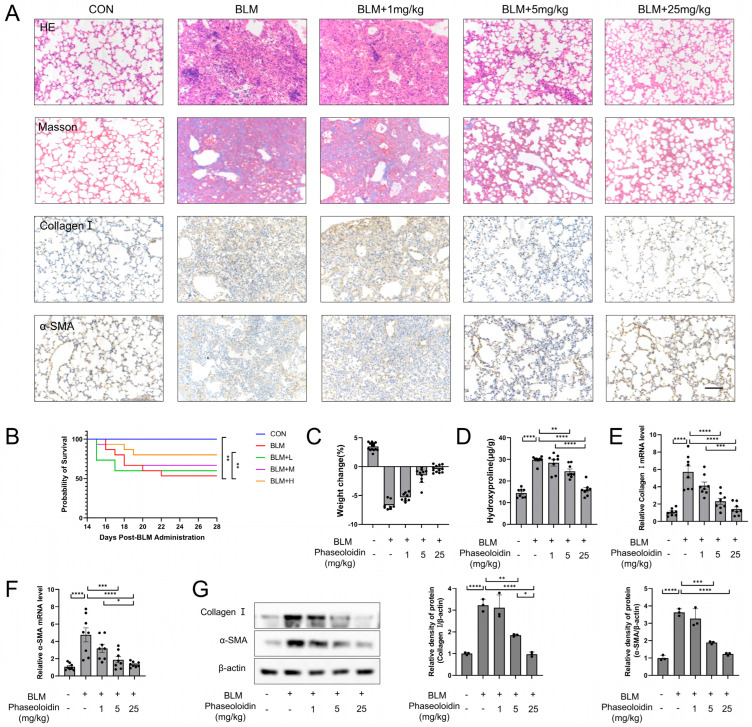
Phaseoloidin reverses the bleomycin-induced established lung fibrosis in mice. (**A**) Bleomycin (5 mg/kg) was given by tracheal injection on day 0, and phaseoloidin (1 mg/kg, 5 mg/kg, 25 mg/kg) was given by intraperitoneal injection from day 14 to day 28. Histology, Masson staining, and immunohistochemistry of Collagen I and α-SMA were analyzed for control, BLM (bleomycin), BLM + 1 mg/kg (bleomycin + 1 mg/kg phaseoloidin), BLM + 5 mg/kg (bleomycin + 5 mg/kg phaseoloidin) and BLM + 25 mg/kg (bleomycin + 25 mg/kg phaseoloidin) groups. (**B**) Survival curves of the above experimental animal groups. (**C**) Weight changes in the above experimental animal groups. (**D**) Hydroxyproline content in the above-mentioned animal experimental groups. (**E**,**F**) Quantification of mRNA expression of Collagen I and α-SMA in the above-mentioned animal experimental groups. (**G**) Western blot analysis of Collagen I and α-SMA levels in experimental animal groups (**left**) and quantification of the band intensity (**right**). Bars represent 100 μm (**A**). Data represent the means ± S.D. (* *p* < 0.05, ** *p* < 0.01, *** *p* < 0.001, **** *p* < 0.0001).

**Figure 2 biomedicines-13-02679-f002:**
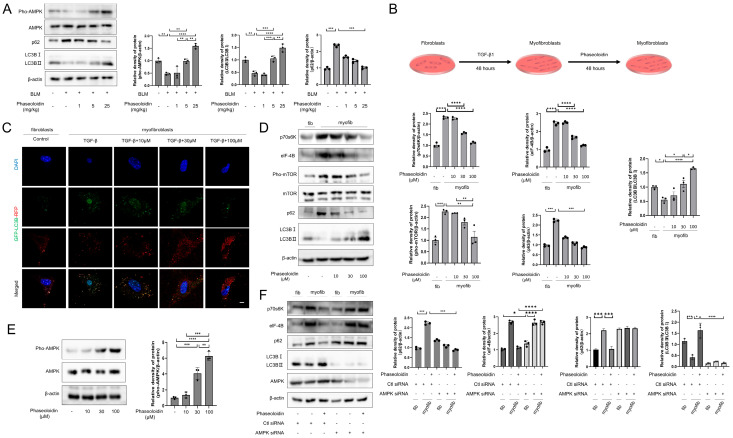
Phaseoloidin promotes autophagy via the AMPK/mTOR pathway. (**A**) Western blot analysis of Pho-AMPK, AMPK, p62 and LC3B in animal groups mentioned above (**left**) and quantification of the band intensity (**right**). (**B**) schematic diagram of the phaseoloidin-treated cell model. Primary mouse fibroblasts became myofibroblasts after induction by TGF-β1 for 48 h, then treated with or without phaseoloidin. (**C**) GFP-LC3B-RFP plasmid transfected for 24 h and observed under APOTOME fluorescence microscope. (**D**) Western blot analysis of Pho-mTOR, mTOR, p70S6K, elf-4B, p62 and LC3B in the groups mentioned above (**left**) and quantification of the band intensity (**right**). (**E**) Western blot analysis of Pho-AMPK and AMPK levels in phaseoloidin-treated primary mouse myofibroblasts (**left**) and quantification of the band intensity (**right**). (**F**) Western blot analysis of p70S6K, elf-4B, p62, LC3B, and AMPK under Ctl siRNA or AMPK siRNA treated (**left**) and quantification of the band intensity (**right**). Bars represent 20 μm (**C**). Data represent the means ± S.D. (* *p* < 0.05, ** *p* < 0.01, *** *p* < 0.001, **** *p* < 0.0001).

**Figure 7 biomedicines-13-02679-f007:**
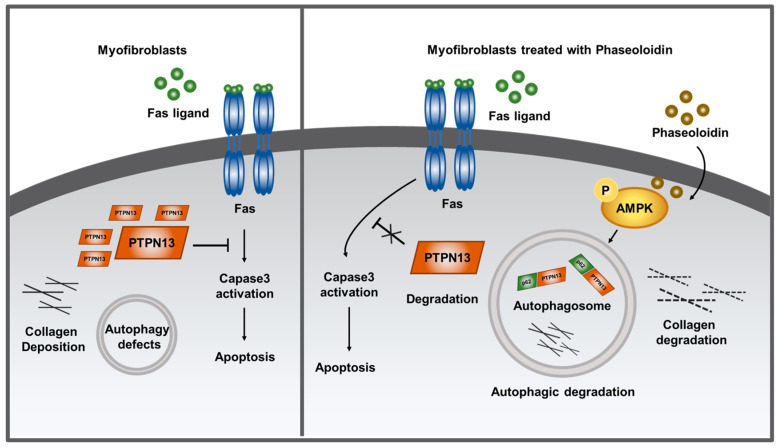
Phaseoloidin activates AMPK and enhances autophagy in myofibroblasts with autophagy defects.

## Data Availability

The datasets generated during and/or analyzed during the current study are available from the corresponding author on reasonable request.

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
