# Peer review of "Reversal of Myofibroblast Apoptosis Resistance and Collagen Deposition by Phaseoloidin-Induced Autophagy Attenuates Pulmonary Fibrosis"

_biomedicines, 2025, doi:10.3390/biomedicines13112679_

Round 1
Reviewer 1 Report
Comments and Suggestions for Authors
-
In Figure 2, you present data using the LC3 antibody. Have you also examined p62 as an additional marker of autophagy in this context? Including p62 data would further strengthen your findings.
-
Your animal model includes Control, Bleomycin, and Bleomycin with Phaseoloidin treatment groups. Did you consider including a group treated with Phaseoloidin alone? Including such a group would clarify the specific effects of Phaseoloidin in the absence of Bleomycin and its impact on the mouse model.
-
The text in Figure 4 is difficult to read. Please improve the figure’s clarity and ensure all labels and text are legible.
-
While you demonstrate the effects of Phaseoloidin in a respiratory disease model, have you explored its efficacy in other lung injury models? Discussion of any additional models or relevant literature would be valuable.
-
Have you considered translational studies in humans? For future directions, do you plan to analyze human lung lysates or conduct related studies to support the clinical relevance of your findings?
-
Please expand the Discussion section to address and compare Figures 1–6 with existing literature. This will help contextualize your results within the broader field.
-
Lastly, please discuss the limitations of your study and potential future directions in the manuscript.
Author Response
Response to Reviewer 1’s comments
Thank you very much for your detailed and insightful comments on our manuscript. We much appreciate your time and contribution to providing constructive feedback on the ways to improve our paper. The main changes have been highlighted in the revised manuscript. To facilitate your review of our revisions, the following is a point-by-point response to the questions and comments you proposed. Thanks again for your assistance and support.
1.In Figure 2, you present data using the LC3 antibody. Have you also examined p62 as an additional marker of autophagy in this context? Including p62 data would further strengthen your findings.
Response:
- We appreciate the reviewer’s constructive suggestion. In addition to LC3, we have also examined p62 as an autophagy marker. The results showed that p62 expression levels were consistent with the LC3 data, further confirming the activation of autophagy. The corresponding data have now been included in Figure 2 of the revised manuscript.
2.Your animal model includes Control, Bleomycin, and Bleomycin with Phaseoloidin treatment groups. Did you consider including a group treated with Phaseoloidin alone? Including such a group would clarify the specific effects of Phaseoloidin in the absence of Bleomycin and its impact on the mouse model.
Response:
-We sincerely appreciate this valuable suggestion. In accordance with your recommendation, we previously included We included a Phaseoloidin-only treatment group as a comparator to isolate the drug-specific effects. The results from this group, however, were not included in the main text and are presented in Supplementary Figure 1. Histological analyses using H&E and Masson’s staining demonstrated that Phaseoloidin alone did not induce pulmonary fibrosis. In addition, hydroxyproline quantification further confirmed that collagen deposition was not increased in the Phaseoloidin alone group compared with controls. These findings collectively indicate that Phaseoloidin itself does not cause adverse effects on lung morphology or fibrosis progression. And the result are described in the revised manuscript. (line222-224)
3.The text in Figure 4 is difficult to read. Please improve the figure’s clarity and ensure all labels and text are legible.
Response:
-We thank the reviewer for this helpful comment. Figure 4 has been revised to improve its clarity, and all labels and text have been reformatted to ensure legibility. The updated figure has been uploaded in the revised manuscript.
4.While you demonstrate the effects of Phaseoloidin in a respiratory disease model, have you explored its efficacy in other lung injury models?
Response:
-We thank the reviewer for this insightful suggestion. In the present study, we focused on evaluating the effects of Phaseoloidin in a respiratory disease model. We have not yet assessed its efficacy in other lung injury models; however, we agree that this is an important direction and we plan to further explore it in our future work. (line509-511)
5.Have you considered translational studies in humans? For future directions, do you plan to analyze human lung lysates or conduct related studies to support the clinical relevance of your findings?
Response:
-We thank the reviewer for this important comment. As Phaseoloidin is a natural compound, our current study was limited to experimental models, and clinical trials are beyond the scope of this work. Nevertheless, we believe our findings provide valuable preclinical evidence and may serve as a foundation for future studies analyzing human lung tissues or clinical samples to support the translational potential of Phaseoloidin. We have added this point to the Discussion section accordingly (lines441-445,).
6.Please expand the Discussion section to address and compare Figures 1–6 with existing literature. This will help contextualize your results within the broader field.
Response:
-We thank the reviewer for this helpful comment. To contextualize our findings within the broader field, we have expanded the Discussion section to address and compare Figures 1–6 with existing literature (line421-436,495-511)
7.Lastly, please discuss the limitations of your study and potential future directions in the manuscript.
Response:
-We thank the reviewer for this helpful comment. We have discussed the limitations of your study and potential future directions in the discussion part of the manuscript. (line437-445,466-469)

Reviewer 2 Report
Comments and Suggestions for Authors
Dear Editor,
Major Comment:
In the original report titled: “Reversal of Myofibroblast Apoptosis Resistance and Collagen Deposition by Phaseoloidin Induced Autophagy Attenuate Pulmonary Fibrosis”, Siyuan Li and colleagues utilize a mouse model of bleomycin-induced pulmonary fibrosis to investigated the therapeutic activity of Phaseoloidin in ameliorating pulmonary fibrosis and the pathways involved in this activity. They demonstrated that this activity involved the promotion of autophagy and clearance of excessive collagen deposition. They show that this was associated with the AMPK/mTOR pathway and the resolution of myofibroblast apoptosis resistance through degrading PTPN13 activity. The manuscript has a sound scientific design and methodology, and addresses an important, clinically relevant question. There were however a number of contradictions in the text. The authors contradict their own data and established knowledge in the literature. Multiple references were incorrectly cited. There were also multiple grammatical errors in the text and illegible figures and figure labels that make understanding the manuscript difficult.
Although the data presented has good basic and translational science merit, the multiple scientific and grammatical errors the manuscript contains makes it unsuitable and unpublishable in its current state. At least a major revision is hereby advised. I list below my specific comments.
- The role of mTOR as a negative regulator of autophagy is not clearly described in the text. This is important for a good understanding of the biological activity of Phaseoloides based on their hypothesis. Furthermore, the authors say in the text that “phaseoloidin was found to induce phosphorylation of mTOR in a concentration gradient, the p70S6K and elf-4B increased followed, thereby promoting autophagy”. This is very confusing and contrary to their figures which shows a decrease in phos-mTOR with increasing concentration of Phaseoloidin. They also state: “Moreover, AMPK siRNA reversed the increase of p70S6K and elf-4B following Phaseoloidin treatment”. This is counter intuitive and contrasts the data presented in the figures. Similarly, the statement in line 391-393 that “phaseoloidin could induce the phosphorylation of mTOR, increase the expression of p70S6K and elf-4B counteracts the results presented and goes against knowledge in the literature. The authors must reconcile these stark discrepancies. They need to clearly describe what phosphorylation of mTOR at specific cites achieve in the AMPK/mTOR pathway.
- The term “Autophagy detects” appeared multiple times in the text including but not limited to lines 42, 48, 370. I believe this should be “autophagy defects” or autophagy-defective as the case may be.
- The authors claim that Phaseoloides is the main active ingredient from Entada phaseoloides and reference a study titled “Study on Analgesic Effect of Dai Medicine Entada phaseoloides Seed Kernel and Its Total Saponins” published in the Journal of Medicine and Pharmacy of Chinese Minorities. Firstly, the title in the journal, though similar, is slightly different from the reference so this needs to be corrected. Given that the said article is not open access and not in English, could the authors provide another complementary reference to back up their statement?
- Materials and Methods section 2.7: The section is difficult to read and understand as the authors switched from describing the procedure they followed to instructing on a procedure to follow. The section is also very poorly punctuated. Overall, the method section can be better punctuated with shorter sentences to improve readability.
- In some part of the text (including but not limited to line 238) the authors refer to multiple studies but include only one reference. The authors should provide more than one references where they have suggested their statement is supported my multiple publications.
- The word ‘restore’, as used frequently in the text confuses the meaning of what is conveyed. For example, in line 288, “Phaseoloidin Restores Apoptosis Resistance of Myofibroblasts to FasL” suggests that Phaseoloidin makes myofibroblasts resistant to apoptosis. For better understanding, the statement could read: “Phaseoloidin Resolves (or Reverses or corrects) Apoptosis Resistance of Myofibroblasts to FasL”. Same applies for multiple lines in the text including but not limited to 328, 432, 450, 210.
- Could the authors explain the variations in the datapoint in Figure 1 graphs? The experiment groups contained 15 mice per group based on their materials and methods but different numbers of data points are presented across figure 1.
- The three bar graphs in Figure 2F each show 2 sets of same experimental conditions with significantly different results. This makes no sense unless the graph is missing further labeling. Same goes for figure 3B, 3C and 3D
- Where ever results are quoted in the text, the authors need to cite the appropriate figures. For example line 207
- Line 200-201: it is not clear whether the authors are referring to the current work or to some previous publication.
- Some figures are poorly described. Authors should describe what the figures show and not just mention the assay that was used (as with their description of Fig 3A) nor what they interpret the results to mean (as with their description of Fig 3C LC3B II/I)
- Flow cytometry dot plot labels in Figure 4B and 4D are not legible. As a whole, Figure 4 is quite blurry. Picture quality should be improved.
- lines 315 - 316 is confusing. Authors should clarify the role of PTPN13 and p62 with regards to apoptosis and provide appropriate references.
- Lines 369-370: Authors should acknowledge the previous report that Entada Phaseoloides extract activates AMPK
- In line 409-410, Bcl-2, XIAP, cFLIPL, and PTPN13 are described at apoptotic genes which is incorrect.
- In line 422, the study by Furuya et al 2005 is incorrectly cited at Pattingre et al.
- Lines 474-427 contains an incorrect citation. The cited publication is a review article by Fitzwalter et al and not original research by Thorburn et al and the authors stated.
- The statement in line 435 – 437 “Therefore, these data suggest that the restoration of apoptosis resistance in myofibroblasts by phaseoloidin is mediated by PTPN13” is contradictory.
- Line 441: reference 51 appears to be a wrong reference for the statement on AMPK activity in IPF patients.
- Line 443: reference 39 is an incorrect reference. It is not a preclinical study on AMPK activators and their protection against lung injury.
- Lines 446 – 448 contains multiple typographical errors
The manuscript contains multiple grammatical errors that makes understanding the text difficult. I have pointed to some of those errors in my suggestions for authors.
Author Response
Response to Reviewer 2’s comments
Thank you very much for your detailed and insightful comments on our manuscript. We much appreciate your time and contribution to providing constructive feedback on the ways to improve our paper. The main changes have been highlighted in the revised manuscript. To facilitate your review of our revisions, the following is a point-by-point response to the questions and comments you proposed. Thanks again for your assistance and support.
Major Comment:
1.In the original report titled: “Reversal of Myofibroblast Apoptosis Resistance and Collagen Deposition by Phaseoloidin Induced Autophagy Attenuate Pulmonary Fibrosis”, Siyuan Li and colleagues utilize a mouse model of bleomycin-induced pulmonary fibrosis to investigated the therapeutic activity of Phaseoloidin in ameliorating pulmonary fibrosis and the pathways involved in this activity. They demonstrated that this activity involved the promotion of autophagy and clearance of excessive collagen deposition. They show that this was associated with the AMPK/mTOR pathway and the resolution of myofibroblast apoptosis resistance through degrading PTPN13 activity. The manuscript has a sound scientific design and methodology, and addresses an important, clinically relevant question. There were however a number of contradictions in the text. The authors contradict their own data and established knowledge in the literature. Multiple references were incorrectly cited. There were also multiple grammatical errors in the text and illegible figures and figure labels that make understanding the manuscript difficult.
Although the data presented has good basic and translational science merit, the multiple scientific and grammatical errors the manuscript contains makes it unsuitable and unpublishable in its current state. At least a major revision is hereby advised. I list below my specific comments.
1.The role of mTOR as a negative regulator of autophagy is not clearly described in the text. This is important for a good understanding of the biological activity of Phaseoloides based on their hypothesis. Furthermore, the authors say in the text that “phaseoloidin was found to induce phosphorylation of mTOR in a concentration gradient, the p70S6K and elf-4B increased followed, thereby promoting autophagy”. This is very confusing and contrary to their figures which shows a decrease in phos-mTOR with increasing concentration of Phaseoloidin. They also state: “Moreover, AMPK siRNA reversed the increase of p70S6K and elf-4B following Phaseoloidin treatment”. This is counter intuitive and contrasts the data presented in the figures. Similarly, the statement in line 391-393 that “phaseoloidin could induce the phosphorylation of mTOR, increase the expression of p70S6K and elf-4B counteracts the results presented and goes against knowledge in the literature. The authors must reconcile these stark discrepancies. They need to clearly describe what phosphorylation of mTOR at specific cites achieve in the AMPK/mTOR pathway.
Response:
We sincerely thank the reviewer for pointing out this important issue. We agree that our description of the role of mTOR in autophagy regulation was not sufficiently clear and that some statements were inconsistent with the data. In our study, Phaseoloidin treatment led to a concentration-dependent decrease in mTOR phosphorylation, consistent with the promotion of autophagy. Accordingly, the phosphorylation of p70S6K and eIF4B was reduced rather than increased, in line with the literature that mTOR acts as a negative regulator of autophagy. We have carefully revised the Results and Discussion sections to correct these discrepancies and to provide a clearer description of the AMPK/mTOR pathway. In particular, we now emphasize that mTOR phosphorylation at specific sites (e.g., Ser2448) reflects its activation status, and Phaseoloidin suppressed this phosphorylation through AMPK activation, thereby promoting autophagy. These revisions reconcile our findings with the figures and current knowledge in the field.
2.The term “Autophagy detects” appeared multiple times in the text including but not limited to lines 42, 48, 370. I believe this should be “autophagy defects” or autophagy-defective as the case may be. (Line52,59,407,420)
Response:
-We thank the reviewer for catching this error. The term “autophagy detects” was indeed a typographical mistake. We have carefully revised the text and corrected it to “autophagy defects” or “autophagy-defective” as appropriate in all instances (lines 52, 59, 407, Line 420 in the revised manuscript).
3.The authors claim that Phaseoloides is the main active ingredient from Entada phaseoloides and reference a study titled “Study on Analgesic Effect of Dai Medicine Entada phaseoloides Seed Kernel and Its Total Saponins” published in the Journal of Medicine and Pharmacy of Chinese Minorities. Firstly, the title in the journal, though similar, is slightly different from the reference so this needs to be corrected. Given that the said article is not open access and not in English, could the authors provide another complementary reference to back up their statement?
Response:
-We thank the reviewer for this careful and constructive comment. As the originally cited article is not open access and is not available in English, it has been removed, and two peer-reviewed references that more directly support the statements regarding Entada phaseoloides and the bioactivity of phaseoloidin have been incorporated into the revised manuscript. These complementary references provide independent pharmacological evidence that phaseoloidin is a major bioactive constituent and document its reported anti-inflammatory and antioxidant activities.
4.Materials and Methods section 2.7: The section is difficult to read and understand as the authors switched from describing the procedure they followed to instructing on a procedure to follow. The section is also very poorly punctuated. Overall, the method section can be better punctuated with shorter sentences to improve readability.
Response:
-We thank the reviewer for this helpful and specific suggestion. Section 2.7 has been rewritten to improve clarity, punctuation, and readability. The revised text (which replaces the original Section 2.7) is given below and has been incorporated into the revised manuscript.(line160-172)
5.In some part of the text (including but not limited to line 238) the authors refer to multiple studies but include only one reference. The authors should provide more than one references where they have suggested their statement is supported my multiple publications.
Response:
Thank you for pointing out this problem. The statements in question are now supported by multiple citations,We appreciate the reviewer’s comment, which has strengthened the manuscript’s evidential support.(line256)
6.The word ‘restore’, as used frequently in the text confuses the meaning of what is conveyed. For example, in line 288, “Phaseoloidin Restores Apoptosis Resistance of Myofibroblasts to FasL” suggests that Phaseoloidin makes myofibroblasts resistant to apoptosis. For better understanding, the statement could read: “Phaseoloidin Resolves (or Reverses or corrects) Apoptosis Resistance of Myofibroblasts to FasL”. Same applies for multiple lines in the text including but not limited to 328, 432, 450, 210.
Response:
-We thank the reviewer for this important suggestion. All instances of the ambiguous term “restore/restored” have been reviewed and revised to eliminate potential misunderstanding. Wherever appropriate, the terms reverse, resolve, or correct have been used. (line24,72,226,309,312,314,484,486)
7.Could the authors explain the variations in the datapoint in Figure 1 graphs? The experiment groups contained 15 mice per group based on their materials and methods but different numbers of data points are presented across figure 1.
Response:
-We thank the reviewer for this careful observation and apologize for any confusion. All groups were initiated with 15 mice; however, bleomycin administration induces mortality in murine pulmonary fibrosis models and the intraperitoneal administration of different doses of phaseoloidin produced dose-dependent effects on survival[1]. Consequently, the number of surviving animals available for each measurement differed between groups, and the number of datapoints shown in the individual panels of Figure 1C,1D,1E and 1F therefore reflects the actual number of animals included in those analyses.
We are grateful to the reviewer for prompting this clarification.
- Peng, R., et al., Bleomycin induces molecular changes directly relevant to idiopathic pulmonary fibrosis: a model for "active" disease. PLoS One, 2013. 8(4): p. e59348.
8.The three bar graphs in Figure 2F each show 2 sets of same experimental conditions with significantly different results. This makes no sense unless the graph is missing further labeling. Same goes for figure 3B, 3C and 3D
Response:
Thank you for this careful and constructive observation. We apologize for the resulting confusion. To avoid ambiguity, the bar graphs have been appropriately modified.
For Figure 2F, the two data sets showing identical nominal treatment labels (e.g. Phaseoloidin–, Ctl siRNA–, AMPK siRNA+) were different experimental contexts: primary mouse fibroblasts versus primary mouse myofibroblasts. The myofibroblasts were derived from fibroblasts after treatment with TGF-β. The corresponding Western-blot panels already show the two sample types, but the bar graphs were not sufficiently annotated to make this clear at a glance. Similarly, for Figures 3B–3D, the panels that appear to share identical condition labels actually represent results from different experimental contexts.
To avoid ambiguity, additional labels have now been included below the bar graphs to clearly indicate whether the data correspond to fibroblasts or myofibroblasts. We thank the reviewer again for highlighting this issue, which helped us improve the clarity of figure presentation.
9.Where ever results are quoted in the text, the authors need to cite the appropriate figures. For example line 207
Response:
Thank you for this valuable comment. All result statements in the manuscript have been systematically anchored to their corresponding figures with the following modifications.
10.Line 200-201: it is not clear whether the authors are referring to the current work or to some previous publication.
Response:
-We thank the reviewer for pointing out this ambiguity and apologize for the lack of clarity. To avoid ambiguity, we have revised the sentence accordingly. This revised sentence clarifies that the statement is based on previous research rather than the current study. (line213-216)
11.Some figures are poorly described. Authors should describe what the figures show and not just mention the assay that was used (as with their description of Fig 3A) nor what they interpret the results to mean (as with their description of Fig 3C LC3B II/I)
Response:
Thank you for this valuable comment. We have revised the descriptions of Figure 3A and Figure 3C in accordance with your suggestions to ensure precise presentation of experimental data, rather than merely stating assay methods or directly interpreting results.
12.Flow cytometry dot plot labels in Figure 4B and 4D are not legible. As a whole, Figure 4 is quite blurry. Picture quality should be improved.
Response:
We thank the reviewer for pointing this out. We have replaced Figure 4 with higher-resolution images, and the labels in Figure 4B and 4D have been adjusted to ensure clarity and legibility.
13.ines 315 - 316 is confusing. Authors should clarify the role of PTPN13 and p62 with regards to apoptosis and provide appropriate references.
Response:
Thank you for this valuable comment. The role of PTPN13 and p62 in regulating apoptosis has been clarified in the revised manuscript. (Lines336-341)
14.Lines 369-370: Authors should acknowledge the previous report that Entada Phaseoloides extract activates AMPK
Response:
Thank you for this valuable comment. We have acknowledged the previous report. (Lines398-401)
15.In line 409-410, Bcl-2, XIAP, cFLIPL, and PTPN13 are described at apoptotic genes which is incorrect.
Response:
We thank you for identifying this inaccuracy and apologize for the imprecise wording.
The phrase “apoptotic genes” was incorrect and has been corrected to accurately reflect the role of these regulators. (lines456-458)
16.In line 422, the study by Furuya et al 2005 is incorrectly cited at Pattingre et al.
Response:
Thank you for identifying this error, and we apologize for the oversight. We have changed it.(lines472)
17.Lines 474-427 contains an incorrect citation. The cited publication is a review article by Fitzwalter et al and not original research by Thorburn et al and the authors stated.
Response:
Thank you for pointing out this error, and we apologize for the oversight. We have changed it.(lines474)
18.The statement in line 435 – 437 “Therefore, these data suggest that the restoration of apoptosis resistance in myofibroblasts by phaseoloidin is mediated by PTPN13” is contradictory.
Response:
Thank you for this valuable comment. We have revised the descriptions of Figure 3A and Figure 3C in accordance with your suggestions to ensure precise presentation of experimental data, rather than merely stating assay methods or directly interpreting results.
19.Line 441: reference 51 appears to be a wrong reference for the statement on AMPK activity in IPF patients.
Response:
Thank you for this helpful comment. Reference 51 has been replaced by a primary study. The newly included references are changed(lines473)
20.Line 443: reference 39 is an incorrect reference. It is not a preclinical study on AMPK activators and their protection against lung injury.
Response:
We thank the reviewer for identifying this error. The incorrect citation has been removed and replaced in the reference list with the following preclinical study, which more appropriately supports the statement regarding AMPK activation and attenuation of lung fibrosis.(lines491)
21.Lines 446 – 448 contains multiple typographical errors
Response:
Thank you for carefully pointing out the typographical issues in lines 446–448. We apologize for the oversight, and the passage has now been corrected and clarified. (lines506-511)

Round 2
Reviewer 2 Report
Comments and Suggestions for Authors
In the revised manuscript, the authors have satisfactorily addressed the majority of the comments I previously raised. However, the claim in the introduction and conclusion, that phaseoloidin in "the major" bioactive component of Entada phaseoloides is unsubstantiated by references or data.
Comments on the Quality of English LanguageThe quality of English is still poor and affects understanding of some areas of the text although there is improvement over the previous version especially in the portions that were previously highlighted.
Author Response
Response to Reviewer 2’s comments
- In the revised manuscript, the authors have satisfactorily addressed the majority of the comments I previously raised. However, the claim in the introduction and conclusion, that phaseoloidin in "the major" component of Entada phaseoloides is unsubstantiated by references or data.
Response:
-Thank you for this important comment. We agree that the statement in the Introduction and Conclusion asserting that “phaseoloidin is the major component of Entada phaseoloides” was too strong as presented without direct citation or quantitative support. To clarify this point, we have included an additional reference to indicate that phaseoloidin is one of the components, rather than the sole major constituent, of E. phaseoloides.
To address this, we have revised the manuscript as follows (Line 74-77 and Line 509-511)
- The quality of English is still poor and affects understanding of some areas of the text although there is improvement over the previous version especially in the portions that were previously highlighted.
Response:
-Thank you for this important comment. We sincerely apologize for the persistent issues with the English quality in our previous submission and We have engaged MDPI Author Services for a comprehensive second-round polishing.

Round 3
Reviewer 2 Report
Comments and Suggestions for Authors
I have no further suggestions as the authors have satisfactorily addressed the issues raised in the previous reviews.